# Study on Adhesion Performance and Aging Strength Degradation Mechanism of SBS Modified Asphalt with Different Anti-Aging Additive

**DOI:** 10.3390/ma16134881

**Published:** 2023-07-07

**Authors:** Chuanyi Zhuang, Hao Guo, Fengxiang Li, Yan Hao, Kun Chen, Gen Li, Yali Ye

**Affiliations:** 1School of Transportation and Civil Engineering, Shandong Jiaotong University, Jinan 250357, China; 204095@sdjtu.edu.cn (C.Z.); 13396446296@163.com (H.G.); chen1921701@163.com (K.C.); li815485689@163.com (G.L.); 204068@sdjtu.edu.cn (Y.Y.); 2Shandong Quality Inspection and Testing Center of Construction Engineering Co., Ltd., Jinan 250100, China; lfx168@126.com

**Keywords:** SBS composite modified asphalt, surface free energy method, adhesion properties, anti-aging agent, lying drip method

## Abstract

After aging, the adhesiveness of asphalt deteriorates, leading to a reduction in the durability of asphalt mixtures and affecting the service life of asphalt pavements. To enhance the anti-aging performance of asphalt, this study employed the method of melt blending to prepare three types of modified asphalt: graphene/SBS modified asphalt (G/SBSMA), crumb rubber/SBS modified asphalt (CR/SBSMA), and petroleum resin/SBS modified asphalt (PR/SBSMA). Different dosages of the three types of modified asphalt were tested for changes in conventional performance indicators. The optimal dosages of graphene, crumb rubber, and C9 petroleum resin were determined to be 2%, 15%, and 5%, respectively. Based on the theory of surface free energy, the effects of anti-aging agents on the microscopic properties of SBS modified asphalt before and after aging were analyzed using the three-liquid method. The mechanisms of strength attenuation at the asphalt–aggregate interface under water exposure and aging were revealed. The results showed that with the increase of aging gradient, the asphalt-aggregate biphasic system became more active. The cohesive energy and peel energy of SBS modified asphalt increased continuously, while the adhesive energy decreased continuously, leading to a decrease in the energy ratio parameter. Resin-based anti-aging agents exhibited the most significant improvement in asphalt adhesion performance, while graphene demonstrated a more stable enhancement in asphalt’s water stability during the aging stage.

## 1. Introduction

Inadequate adhesive between asphalt and aggregate and poor aging resistance of asphalt can readily result in cracks, looseness, potholes, and other asphalt pavement diseases. Therefore, improving the adhesive between asphalt and aggregate and enhancing the long-term aging resistance of asphalt are the goals pursued by road workers. These goals are also core elements related to the long-term durability of road surfaces [1]. Aging modifiers are frequently utilized in asphalt and its mixtures to increase pavement longevity and maintain high service levels [2]. Among them, nano graphene, rubber powder, petroleum resin, and SBS modified asphalt have good compatibility and excellent performance. They are widely used modified asphalt materials.

The unique layered structure of graphene endows it with extremely strong surface activity, which can improve the mechanical strength of asphalt mixtures when physically blended with asphalt. It can also hinder the diffusion and penetration of oxygen in asphalt, potentiate thermal stability and barrier performance, and achieve the goal of delaying aging. Li [3] pointed out that a 1% content of graphene can reduce the temperature receptivity of the base asphalt and improve the rheological properties of the asphalt before and after aging. Liu [4] proposed that a 0.4% content of graphene could reduce the phase angle of SBS modified asphalt and increase its resistance to permanent deformation. Additionally, graphene can reduce the rate of change of the carbonyl index and sulfoxide index after PAV aging. Dong [5] added 2%, 3%, and 4% graphene into the base asphalt. Using the linear amplitude scanning (LAS) test, it was pointed out that the aging resistance of the modified asphalt can be improved when the content of graphene is not more than 3%. The aging resistance effect is more pronounced under the condition of a large strain. Zeng [6] established the molecular model of the asphalt system with graphite oxide and confirmed that graphite oxide can hinder the movement and volatilization of saturated hydrocarbons to a certain extent, thus enhancing the aging resistance of asphalt, based on the density functional theory (DFT). Wang [7] also pointed out that graphite oxide in favor of the crystallization of SBSMA and changes in oxygen containing groups have a positive function on the aging resistance and fatigue endurance of asphalt. Liu [8] highlighted that graphite oxide can not only improve the water stability of the mixture after aging, but also slow down the damage of low-temperature performance.

Waste tire rubber powder, abbreviated as rubber powder, has been widely used in asphalt mixtures to improve the durability of asphalt pavement [9]. However, the modification of rubber powder with asphalt needs to be carried out at higher temperatures, and the environmental impact of the fumes generated during the process cannot be ignored [10,11]. Suo [12] mixed 40-mesh rubber powder with 5%, 10%, 15%, and 20% of SBS modifier and found that the anti-aging effect of composite modified asphalt was better than that of SBS modified asphalt. Jamal [13] modified the base asphalt with 30-mesh, 22.5% high content rubber powder. After laboratory simulated long-term aging and UV aging tests, it was found that the aging index did not change significantly. Using PG grading indicators, Li [14] confirmed that an appropriate amount of rubber powder can improve the low-temperature performance and fatigue performance of asphalt after aging. Xiang [15] studied the micro composition and structural changes of 40 to 60 mesh rubber powder and SBS, before and after thermal oxygen aging, and pointed out that the degradation products of the modifier after aging reacts with the secondary components of asphalt, which will lead to a decrease in aromatic and resin content and an increase in asphaltene content. The increase in the number of macromolecules will contribute to improving micro and macro mechanical properties. Ren [16] established a molecular model of rubber powder modified asphalt using molecular dynamics. Rubber powder can improve the flowability of asphalt molecules, thereby reducing the influence of aging on the low-temperature performance of asphalt.

Modified asphalt with high viscosity has excellent adhesion and elastic recovery ability. The modifiers used to prepare high viscosity asphalt are not unique. Currently, they are mainly divided into rubber, resin, and thermoplastic elastomer types, all of which can be added separately or in a composite form to increase the viscosity of asphalt in order to achieve the standard of high viscosity asphalt with a dynamic viscosity of no less than 20,000 Pa·s at 60 °C [17,18]. Sun [19] found that the phase distribution of asphalt with high viscosity is more uniform and is little affected by the combined effects of oxidation induced hardening and degradation induced softening, demonstrating better aging resistance. Qiu [20] proposed that rubber-based high viscosity agents have the best storage stability and temperature sensitivity through the microscopic and macroscopic indicators of asphalt under different aging states. Shi [21] used C9 petroleum resin and SBS to prepare high viscosity and high elasticity asphalt modifier SPR and found that the economic benefit and anti-aging effect were better. Yang [22] pointed out that aging reduces the strength of functional groups of high viscosity asphalt, but the aging resistance is significantly better than that of base asphalt and SBS modified asphalt.

Currently, research on the anti-aging of asphalt mainly focuses on macroscopic properties, such as high temperature, low temperature, water stability, and fatigue, with less emphasis on the study of microcosmic adhesion strength degradation mechanisms before and after aging. Additionally, due to the differences in material characteristics, different anti-aging agents exhibit varying performance improvements when compounded with SBS, and the degradation patterns of adhesion performance are inconsistent. Therefore, this paper selects three popular anti-aging modifiers, namely, nano graphene, rubber powder, and C9 petroleum resin, and uses the melt blending process to mix them with SBS modified asphalt to prepare composite modified asphalt. The adhesion index of the asphalt and aggregate interface is assessed using the lying drop method based on the surface free energy theory. Quantitative analysis is performed on the variations in adhesion performance between asphalt and aggregate before and after aging, revealing the mechanism of asphalt-aggregate adhesion degradation and strength failure after aging. By studying the effect of different anti-aging agents on the adhesion performance of asphalt, it provides a basis for the comparison and selection of asphalt binders such as asphalt pavement, bridge deck pavement, and ultra-thin overlay. The research methodology is shown in Figure 1.

## 2. Materials and Methods

### 2.1. Raw Materials

#### 2.1.1. Asphalt

According to the ‘Standard Test Methods of Bitumen and Bituminous Mixtures for Highway Engineering’ (JTG E20-2011) [23], the performance indexes of SBS modified asphalt (SBSMA) (as shown in Figure 2) were tested. The data obtained are shown in Table 1.

#### 2.1.2. Modifiers

(1)Graphene

Graphene is made of multilayer graphene powder prepared by Suzhou Carbon Graphene Technology Co., Ltd., as shown in Figure 2, and its basic parameters are shown in Table 2.

(2)Rubber powder

The test used Qingdao Green Leaf Technology Development Co., Ltd. 40 mesh rubber powder (Figure 3). The main technical indicators are shown in Table 3.

(3)Petroleum resin

C9 petroleum resin (Figure 4) is produced by petroleum cracking and has good compatibility with asphalt. The main parameters are shown in Table 4.

### 2.2. Preparation and Performance Evaluation of Composite Modified Asphalt

#### 2.2.1. Composite Modified Asphalt Preparation Process

With the purpose of giving full play to the modification effect of the three anti-aging agents from the perspective of the shear process and economy, three kinds of composite-modified asphalts were prepared by the melt blending method.

(1)Graphene/SBS composite modified asphalt (G/SBSMA)

The G/SBSMA was prepared by adding 0.5%, 1.0%, 1.5%, 2.0%, and 2.5% graphene additive [24]. The temperature was set at 170 °C, and graphene was added to asphalt at a rate of 2k rpm in a small number, multiple times, within 30 min. The G/SBSMA composite modified asphalt was obtained by adjusting the speed to 3k rpm and shearing for 30 min.

(2)Crumb rubber/SBS composite modified asphalt (CR/SBSMA)

The blending mass ratios were set to 10%, 12.5%, 15%, 17.5%, and 20%, respectively, and the powder was dried in advance [25]. The rubber powder was added to the asphalt in a small amount, many times, at 180 °C shear temperature and a 1.5k rpm shear rate for 30 min. The shear temperature was kept constant, and the shear rate was adjusted to 4k rpm for 60 min. After the shear was completed, the asphalt was placed in an oven at 170 °C for 30 min to swell and develop; finally, CR/SBSMA composite modified asphalt was prepared.

(3)Petroleum resin/SBS composite modified asphalt (PR/SBSMA)

The mixing amount was 3%, 4%, 5%, 6%, and 7% [22]. At the shear temperature of 170 °C and the shear rate of 2k rpm, C9 petroleum resin was added in a small amount, multiple times, for 30 min. PR/SBSMA composite modified asphalt was prepared by increasing the shear rate to 4k rpm and high-speed shearing for 30 min to fully combine the resin and asphalt.

#### 2.2.2. Asphalt Aging Test

The short-term aging was carried out according to the T0610 asphalt rotary film heating test in the ‘Standard Test Methods of Bitumen and Bituminous Mixtures for Highway Engineering’ (JTG E20-2011) [23]. The test selected an SDY-3061 (85) asphalt rotating film oven. After the asphalt was stirred evenly, 35 ± 0.5 g was weighed in the bottle and rotated for 85 min to complete the short-term aging test.

The long-term aging test was carried out according to the accelerated asphalt aging test of the T0630 pressure aging instrument in the test procedure. The Prentex 9300 PAV pressure aging instrument was selected to obtain the long-term aging sample. The setting temperature was 100 °C, with air pressure 2.1 ± 10 min.

### 2.3. Surface Free Energy Methods

In terms of evaluating the adhesion between asphalt and aggregates, China’s ‘Standard Test Methods of Bitumen and Bituminous Mixtures for Highway Engineering’ (JTG E20-2011) [23] recommends the use of the water boiling method and the water immersion method to evaluate. Although the water boiling method and the water immersion method have the advantages of simple operation and intuitive test results, the determination of adhesion level requires manual visual inspection, which has significant errors, and the test results cannot quantitatively characterize the interfacial adhesion performance between asphalt and aggregate [26].

Surface free energy (SFE) is the value of energy required to create a unit area of the new surface in the vacuum condition [27,28]. When the liquid is in contact with the solid, the diffusion phenomenon occurs spontaneously, thereby reducing the free energy of the solid surface to form a stable interface [29]. The angle is formed based on the balance of cohesion between liquids and adhesion between solid and liquid and is called the contact angle (*θ*). The three-phase equilibrium relationship of solid, liquid, and gas is also known as the famous Young equation (T.Young equation):(1)$$\Gamma_{L}\cos\theta =\Gamma_{S}-\Gamma_{SL}$$
where Γ*_L_* is liquid surface tension, Γ*_S_* is solid surface energy, Γ*_SL_* is solid–liquid interface surface energy. The measurement principle of the contact angle is shown in Figure 5.

Surface energy is composed of the polar component (Γ*^AB^*) and non-polar component (Lifshitz-van der Waals component, Γ*^LW^*). The polar components include Lewis acid (Γ*^+^*) and Lewis base (Γ*^−^*). The surface–energy relationship is shown in Formula (2):(2)$$\Gamma =\Gamma^{LW}+\Gamma^{AB}=\Gamma^{LW}+2\sqrt{\Gamma^{+}\Gamma^{-}}$$
where Γ is material surface free energy, Γ*^LW^* is nonpolar component, Γ*^AB^* is polarity component, Γ^+^ is Lewis acid component, Γ^−^ is Lewis base component.

When the solid and liquid are in contact, in order to achieve the solid–liquid equilibrium state, the interface between the two is bound to cause changes in the surface area of the solid and liquid. The interface surface energy can be divided into non-polar components and polar components. The expression of each component change is shown in Formulas (3) and (4):(3)$$\Gamma_{SL}^{LW}=\Gamma_{S}^{LW}+\Gamma_{L}^{LW}-2\sqrt{\Gamma_{S}^{LW}\Gamma_{L}^{LW}}$$
(4)$$\Gamma_{SL}^{AB}=2\left(\sqrt{\Gamma_{S}^{+}\Gamma_{S}^{-}}+\sqrt{\Gamma_{L}^{+}\Gamma_{L}^{-}}-\sqrt{\Gamma_{S}^{+}\Gamma_{L}^{-}}-\sqrt{\Gamma_{L}^{+}\Gamma_{S}^{-}}\right)$$

The surface energy expression of solid-liquid phase can be obtained by adding Formulas (3) and (4):(5)$$\Gamma_{SL}=\Gamma_{S}+\Gamma_{L}-2\sqrt{\Gamma_{S}^{LW}\Gamma_{L}^{LW}}-2\sqrt{\Gamma_{S}^{+}\Gamma_{L}^{-}}-2\sqrt{\Gamma_{S}^{-}\Gamma_{L}^{+}}$$
where Γ*_S_* is solid surface energy; $\Gamma_{\;S}^{LW}$ is solid nonpolar component; $\Gamma_{S}^{+}$, $\Gamma_{S}^{-}$ are solid polar acid, alkali component; Γ*_L_* is liquid surface tension; $\Gamma_{L}^{LW}$ is nonpolar component of liquid; $\Gamma_{L}^{+}$, $\Gamma_{L}^{-}$ are liquid polar acid, alkali component; Γ*_SL_* is the solid–liquid interface surface energy; $\Gamma_{SL}^{LW}$ is the non-polar component of solid–liquid interface; $\Gamma_{SL}^{AB}$ is the solid–liquid interface polarity component.

The relationship between the contact angle and solid-liquid surface energy can be obtained by combining Formulas (1) and (5):(6)$$\Gamma_{L}\left(1+\cos\theta \right)=2\left(\sqrt{\Gamma_{S}^{LW}\Gamma_{L}^{LW}}+\sqrt{\Gamma_{S}^{+}\Gamma_{L}^{-}}+\sqrt{\Gamma_{S}^{-}\Gamma_{L}^{+}}\right)$$

In order to obtain the solid surface energy, it is necessary to solve the three unknown parameters of $\Gamma_{\;S}^{LW}$, $\Gamma_{S}^{+}$, and $\Gamma_{S}^{-}$ and set them to *x*, *y*, and *z*, respectively. The equations can be listed as follows:(7)$$\left[\begin{array}{ccc} \sqrt{\Gamma_{L1}^{LW}} & \sqrt{\Gamma_{L1}^{-}} & \sqrt{\Gamma_{L1}^{+}}\\ \sqrt{\Gamma_{L2}^{LW}} & \sqrt{\Gamma_{L2}^{-}} & \sqrt{\Gamma_{L2}^{+}}\\ \sqrt{\Gamma_{L3}^{LW}} & \sqrt{\Gamma_{L3}^{-}} & \sqrt{\Gamma_{L3}^{+}} \end{array}\right]\left[\begin{array}{c} x\\ y\\ z \end{array}\right]=\frac{1}{2}\left[\begin{array}{c} \left(1+\cos\theta_{1}\right)\cdot \Gamma_{L1}\\ \left(1+\cos\theta_{2}\right)\cdot \Gamma_{L2}\\ \left(1+\cos\theta_{3}\right)\cdot \Gamma_{L3} \end{array}\right]$$

The three equations are used to establish the matrix equation, and the three-liquid method is used to test the three test liquids with different surface energy; finally, the solid surface energy parameters can be obtained.

In order to facilitate the use of surface energy parameters to calculate the adhesion performance index between aggregate and asphalt, the subscripts *A*, *S*, *L*, and *W* below represent asphalt, limestone aggregate, test liquid, and water, respectively.

(1)Cohesion energy

If the homogeneous asphalt phase is separated to produce two new interfaces, the work completed by the asphalt to overcome the intermolecular force in this process is called cohesion energy. It is twice the surface tension (surface free energy) of asphalt. An asphalt cohesion energy model is shown in Formula (8):(8)$$W_{AA}=2\left(\Gamma_{A}^{LW}+\Gamma_{A}^{AB}\right)=2\left(\Gamma_{A}^{LW}+2\sqrt{\Gamma_{A}^{+}\Gamma_{A}^{-}}\right)$$
where *W_AA_* is the asphalt cohesion energy, $\Gamma_{A}^{LW}$ is the asphalt non-polar component, $\Gamma_{A}^{AB}$ is the asphalt polarity component, $\Gamma_{A}^{+}$, $\Gamma_{A}^{-}$ are acid and alkali components of the polar part of asphalt.

(2)Adhesion work

In a dry environment, asphalt is wrapped on the surface of the aggregate, and the work conducted by separating the asphalt from the stable solid–liquid interface is called adhesion work, which is also an important indicator of the degree of wetting between solid and liquid. The adhesion work can be expressed as Formula (9):(9)$$W_{AS}=\Gamma_{A}+\Gamma_{S}-\Gamma_{AS}$$

According to Formula (5), asphalt and limestone are substituted into the solid-liquid surface energy formula, and the asphalt–aggregate interface surface energy formula can be expressed as Formula (10):(10)$$\Gamma_{AS}=\Gamma_{A}+\Gamma_{S}-2\sqrt{\Gamma_{A}^{LW}\Gamma_{S}^{LW}}-2\sqrt{\Gamma_{A}^{+}\Gamma_{S}^{-}}-2\sqrt{\Gamma_{A}^{-}\Gamma_{S}^{+}}$$

The adhesion work model between asphalt and aggregate can be expressed as Formula (11):(11)$$W_{AS}=2\left(\sqrt{\Gamma_{A}^{LW}\Gamma_{S}^{LW}}+\sqrt{\Gamma_{A}^{+}\Gamma_{S}^{-}}+\sqrt{\Gamma_{A}^{-}\Gamma_{S}^{+}}\right)$$
where *W_AS_* is asphalt adhesion work, Γ*_A_* is asphalt surface energy, Γ*_S_* is the surface energy of aggregate, Γ*_AS_* is asphalt–aggregate interface surface energy, $\Gamma_{S}^{LW}$ is nonpolar component of aggregate, $\Gamma_{S}^{+}$, $\Gamma_{S}^{-}$ are acid and alkali components of the polar part of the aggregate.

(3)Exfoliated work

In the presence of water, the water molecules replace the asphalt film on the surface of the aggregate, and the asphalt is stripped from the three-phase equilibrium system of asphalt, water, and aggregate to form the work completed by the two new interfaces of aggregate–water and asphalt–water. It is called exfoliated work, which is negative, and its expression is Formula (12):(12)$$W_{AWS}=\Gamma_{AW}+\Gamma_{SW}-\Gamma_{AS}$$

Combined with the solid-liquid surface energy formula, the asphalt exfoliated work model can be expressed in detail as Formula (13):(13)$$W_{AWS}=2\left[\begin{array}{l} \sqrt{\Gamma_{A}^{LW}\Gamma_{W}^{LW}}+\sqrt{\Gamma_{S}^{LW}\Gamma_{W}^{LW}}-\sqrt{\Gamma_{A}^{LW}\Gamma_{S}^{LW}}-\Gamma_{W}\\ +\sqrt{\Gamma_{W}^{+}}\left(\sqrt{\Gamma_{A}^{-}}+\sqrt{\Gamma_{S}^{-}}\right)+\sqrt{\Gamma_{A}^{+}}\left(\sqrt{\Gamma_{W}^{-}}-\sqrt{\Gamma_{S}^{-}}\right)\\ +\sqrt{\Gamma_{S}^{+}}\left(\sqrt{\Gamma_{W}^{-}}-\sqrt{\Gamma_{A}^{-}}\right) \end{array}\right]$$
where *W_AWS_* is the asphalt exfoliated work, Γ*_AW_* is the asphalt–water interface surface energy, Γ*_SW_* is the aggregate–water interface surface energy, $\Gamma_{W}^{LW}$ is the water nonpolar component, $\Gamma_{W}^{+}$, $\Gamma_{W}^{-}$ are the acid and alkali components of the water polar part.

(4)Energy ratio

In order to enrich the microscopic index of asphalt water stability performance and make up for the limitations of the evaluation methods of cohesion energy, adhesion work, and exfoliated work, the energy ratio parameters *ER*_1_ and *ER*_2_ are introduced to characterize the water damage resistance of asphalt:(14)$$ER_{1}=\left|\frac{W_{AS}}{W_{AWS}}\right|$$
(15)$$ER_{2}=\left|\frac{W_{AS}-W_{A}{}_{A}}{W_{AWS}}\right|$$

In this test, formamide, ethylene glycol, and distilled water were selected to participate in the calculation of asphalt surface energy. The polar component and non-polar component parameters are shown in Table 5 below.

Limestone aggregate was selected to participate in the subsequent analysis of water stability performance index of aggregate-asphalt system. The selected surface energy parameters of limestone are shown in Table 6.

## 3. Results and Analysis

### 3.1. Conventional Performance Evaluation

The experimental results in this section were conducted in accordance with the conventional performance technical requirements of asphalt specified in the “Technical Specifications for Construction of Highway Asphalt Pavement” (JTG F40-2004) [30].

(1)Conventional properties of graphene/SBS composite modified asphalt

The graphene/SBS composite modified asphalt that was prepared according to the asphalt shear step and the basic index test data of the original sample are shown in Table 7.

As the graphene content increases, the softening point and viscosity of G/SBSMA gradually increase. However, the rate of increase slows down after 2% content. Graphene can be uniformly dispersed in SBS modified asphalt, and its layered structure effectively hinders the relative movement of asphalt molecules, thereby increasing viscosity and improving temperature sensitivity and high-temperature performance. With the addition of graphene, the hardening effect of SBS modified asphalt becomes more pronounced, leading to a decrease in the penetration index. This indicates that the addition of graphene affects the viscosity and flowability of the asphalt. Furthermore, the appropriate addition of graphene can improve the low-temperature cracking resistance of the asphalt, as observed from the 5 °C ductility index. The optimal graphene content for low-temperature performance is found to be 1.5% and 2%. However, adding graphene beyond 2% content results in G/SBSMA exhibiting lower ductility than SBSMA, indicating adverse effects on low-temperature performance.

Considering the significant impact of asphalt aging damage on low-temperature performance and the optimal low-temperature performance achieved at a 2% graphene content, along with improvements in softening point, penetration index, and viscosity that are only slightly inferior to those at a 2.5% content, it is reasonable to select 2% as the optimal graphene content.

(2)Conventional performance of crumb rubber/SBS composite modified asphalt

The test results of the original index of rubber powder and SBS composite modified asphalt with different dosages are shown in Table 8.

The table shows that the addition of rubber powder has the most significant impact on the penetration index and viscosity of CR/SBSMA. The viscosity index exhibits the most noticeable change with a 49.7% increase at a 15% content. The addition of rubber powder increases the elastic component of CR/SBSMA, resulting in an initial increase and subsequent decrease in the penetration index. At 17.5% and 20% content, the values are lower than that of SBSMA, indicating a significant reduction in interlayer shear resistance. The ductility shows an initial trend of increasing and then decreasing, with optimal low-temperature performance observed at 15% and 17.5% content. Due to the characteristics of rubber powder, its contribution to the softening point is smaller than that of the SBS modifier. At low content, the addition of rubber powder can lower the softening point of CR/SBSMA. As the proportion of rubber powder increases, more heat is required for rubber softening, resulting in a gradual increase in the softening point value. However, a plateau period occurs between 15% and 20% where the growth rate slows down. 

Considering the comprehensive impact of 15% rubber powder content on conventional performance indicators, further increasing the content would decrease shear resistance, impair workability during construction, and offer limited contribution to the softening point. Therefore, 15% is chosen as the optimal content of rubber powder for subsequent research.

(3)Conventional performance of petroleum resin/SBS composite modified asphalt

The test results of the original index of high viscosity agent and SBS composite modified asphalt with different dosages are shown in Table 9.

Petroleum resin materials exhibit good compatibility with asphalt, and their blending results in excellent thickening effects. After the addition of C9 petroleum resin, the shear process promotes the bonding of double bonds through oxygen, ultimately forming an intertwined network structure of conjugated pi bonds with the SBS modifier and base asphalt [31]. This dense network structure reduces the flowability of the asphalt, leading to a significant increase in the viscosity and softening point of PR/SBSMA, while the penetration index continues to decrease. The softening point shows a noticeable increase within the range of 3% to 5%, but the improvement in high-temperature performance slows down beyond a 5% content. Within the range of 3% to 7% content, the viscosity keeps rising, but the viscosity at 135 °C remains below 3 Pa·s. This indicates that the addition of an appropriate amount of petroleum resin does not affect the workability and usability of the asphalt. The gel-like structure formed by petroleum resin and SBS modified asphalt enhances the adhesiveness, viscoelastic properties, and improves the low-temperature ductility of PR/SBSMA. The ductility shows an initial increasing and then decreasing trend as the content of petroleum resin increases. The maximum ductility is achieved at a 5% content, with an increase of 5.9 cm compared to 0% content.

Considering the optimal low-temperature performance achieved at a 5% content, as well as the balanced improvements in softening point, penetration index, and viscosity, 5% C9 petroleum resin is chosen as the optimal content.

### 3.2. Contact Angle

From Table 10, the contact angle experiments revealed that both asphalt and water exhibited contact angles greater than 90°. This side validation confirms that asphalt is a hydrophobic material, as water is unable to wet the surface of asphalt. Due to the higher proportion of nonpolar components in their surface energy, formamide and ethylene glycol liquids exhibit smaller contact angles with asphalt, indicating a favorable wetting effect. This further confirms that asphalt is mainly composed of nonpolar components and a smaller fraction of polar components. A contact angle test and baseline setting are shown in Figure 6.

According to the measurement of the contact angle, with Γ*_L_* as the horizontal axis and Γ*_L_* cos*θ* as the vertical axis, a fitting analysis was conducted on the contact angle data of three test liquids with the original, short-term aging, and long-term aging asphalt samples. The fitting results are shown in Figure 7.

The correlation coefficients (R^2^) of the fitting results for the contact angles of the four modified asphalts under three different conditions were all greater than 0.93. This demonstrated a strong correlation between Γ*_L_* and Γ*_L_* cos*θ*, further confirming the validity of the contact angle data. 

As the aging process progresses, the aromatic fractions in asphalt undergo transformation into resins, while resins transform into asphaltene at a slower rate [32]. This results in an increase in the resin content and a decrease in the aromatic fraction content within the asphalt system, leading to an overall increase in the surface energy of the asphalt. Petroleum resins, characterized by abundant aromatic compounds, have the highest content of resins and aromatic fractions. Consequently, aged asphalt exhibits the highest surface energy due to the increased content of resins and aromatic fractions. The aging process alters the chemical composition of asphalt and the content of polar functional groups, leading to the formation of larger molecular substances. As a result, the nonpolar components of asphalt gradually increase while the polar components decrease. This weakening of polar functional groups results in reduced water molecule adsorption capacity.

Asphalt is predominantly composed of nonpolar components, and a higher proportion of nonpolar components indicates a stronger physical adhesion capability of asphalt. This leads to increased stability of the two-phase system formed by asphalt and aggregates. A smaller proportion of polar components indicates that asphalt molecules are less prone to water infiltration, thereby better preventing aggregate detachment. From Figure 8, it can be observed that G/SBSMA has the highest proportion of nonpolar components, followed by PR/SBSMA. The addition of rubber powder in SBS modified asphalt, specifically the unsaturated polar rubber CR, results in the highest proportion of polar components and the lowest proportion of nonpolar components in CR/SBSMA. This leads to an unstable asphalt-aggregate system, making it more susceptible to water intrusion.

### 3.3. Cohesion Energy

The cohesive energy within asphalt characterizes its adhesive properties. A higher cohesive energy indicates that asphalt is less prone to cohesive failure in a stable state and possesses a stronger ability to resist deformation and relative displacement caused by external forces.

From Figure 9, it can be observed that under the as-received condition, G/SBSMA exhibited a 3.0% increase in cohesive energy, CR/SBSMA exhibited a 3.1% increase, and PR/SBSMA exhibited a significant increase of 22.7%. The high-viscosity asphalt phase exhibited the most stable phase structure and the most significant improvement in cohesive energy. Cohesive energy is directly related to asphalt viscosity, with higher viscosity indicating a lower likelihood of cohesive failure. Furthermore, cohesive energy gradually increases with the deepening of aging, indicating an enhancement of internal bonding strength within the asphalt. After short-term aging, SBSMA, G/SBSMA, CR/SBSMA, and PR/SBSMA exhibited increases of 2.5%, 1.1%, 1.8%, and 1.2%, respectively, in cohesive energy. In comparison, the increases in cohesive energy after long-term aging were 2.4%, 2.8%, 3.0%, and 2.9%, respectively. Long-term aging appears to have a greater effect on cohesive energy. Excessive cohesive energy after aging can weaken the asphalt’s ability to envelop aggregates and results in adverse effects on its resistance to water damage.

### 3.4. Adhesion Work

The energy released when the asphalt and aggregate are separated to form a new interface per unit area in a dry environment is called adhesion work. A higher adhesive work indicates a greater resistance of asphalt and aggregates to separate, resulting in a more stable system that is less susceptible to water damage phenomena.

From Figure 10, it can be observed that all four types of asphalt exhibit significant adhesive work with limestone, and all three anti-aging agents have played a positive role in increasing the adhesive work. G/SBSMA and PR/SBSMA have a higher proportion of nonpolar components, which enhances their adhesive properties with limestone due to the rich functional group structure [33]. The adhesive work increased by 5.8% and 14.2%, respectively. As aging progresses, the content of aromatic fractions and resins in asphalt gradually decreases, while the asphaltene content increases, resulting in reduced asphalt ductility and weakened adhesion. Consequently, all four types of asphalt exhibit a decreasing trend in adhesive work. After short-term aging, the adhesive work of SBSMA, G/SBSMA, CR/SBSMA, and PR/SBSMA with limestone decreased by 1.3%, 0.8%, 0.6%, and 1.0%, respectively. Compared to short-term aging, the adhesive work reduction after long-term aging was 2.5%, 2.4%, 1.8%, and 2.3%, respectively. Long-term aging has a greater adverse impact on adhesion.

### 3.5. Exfoliated Work

The exfoliated work represents the energy required to displace the asphalt-aggregate system with an asphalt–water or aggregate–water interface per unit area. A higher exfoliated work indicates that asphalt is more susceptible to displacement by water molecules, leading to the detachment of the asphalt-aggregate system and unfavorable effects on water stability. The exfoliated work is calculated as a negative value based on surface energy calculations. To provide a more intuitive comparison, the absolute values of the exfoliated work are used.

From Figure 11, it can be observed that the exfoliated work of asphalt increases after aging, indicating that aging behavior makes asphalt more prone to detachment from the aggregate surface. With the deepening of the aging gradient, the rate of increase in exfoliated work for SBS modified asphalt gradually increases, and the difference between SBS modified asphalt and the other three types of asphalt also increases. This demonstrates that the three anti-aging agents can mitigate asphalt detachment and potentiate the stability of the asphalt-aggregate system. Anti-aging agents can mitigate the damage caused by aging on asphalt by improving the exfoliated work. However, the contribution of rubber powder to the exfoliated work improvement is the smallest, and its effectiveness is inferior to that of graphene and petroleum resin. After short-term aging, the exfoliated work of SBSMA, G/SBSMA, CR/SBSMA, and PR/SBSMA with limestone increased by 2.8%, 1.7%, 4.3%, and 4.6%, respectively. Long-term aging increased the growth rate of exfoliated work by 3.9%, 2.5%, 1.8%, and 1.0%, respectively. Short-term aging is more detrimental to the exfoliated work of CR/SBSMA and PR/SBSMA. Long-term aging is more detrimental to the exfoliated work of SBSMA and G/SBSMA.

### 3.6. Energy Ratio

Energy ratio, also known as the compatibility ratio, represents the wetting compatibility of the asphalt-aggregate system. A higher energy ratio indicates better fusion of asphalt and better resistance to water damage.

From Figure 12, it can be observed that before and after aging, PR/SBSMA exhibited significantly better ER1 performance compared to the other three types of asphalt due to its excellent adhesion and resistance to flake. However, its high cohesive energy made it less prone to aggregate adhesion, resulting in a lower *ER*_2_ value compared to G/SBSMA. G/SBSMA and CR/SBSMA also had a small improvement in the water stability of asphalt, both *ER*_1_ and *ER*_2_, which were better than SBSMA. G/SBSMA had the most balanced improvement in the performance of the asphalt-aggregate system with the largest proportion of nonpolar components, resulting in a greater increase in cohesion energy, adhesion work, and exfoliated work, and the best *ER*_2_ index.

Considering the adverse effects of asphalt aging on water stability, the *ER*_1_ indicators are ranked PR/SBSMA > G/SBSMA > CR/SBSMA > SBSMA and the *ER*_2_ indicators are ranked G/SBSMA > PR/SBSMA > CR/SBSMA > SBSMA.

## 4. Conclusions

Based on the theory of surface free energy, an analysis was conducted on the impact of three types of anti-aging agents on the conventional performance and microscale indicators (cohesive energy, adhesive work, exfoliated work, and energy ratio parameters) of SBS modified asphalt. The main conclusions were as follows:

(1) The viscosity of the three types of composite modified asphalt that were prepared through the melt blending method increased with the increase of dosage. Due to the excellent viscosity-enhancing effect of C9 petroleum resin itself, the viscosity increased to a greater extent compared to graphene and crumb rubber. Simultaneously, petroleum resin exhibited the most significant improvement in the low-temperature performance at 5 °C, followed by crumb rubber. A moderate amount of graphene also enhanced the low-temperature crack resistance performance.

(2) By considering the fundamental performance indicators of composite modified asphalt before and after aging, the optimal dosage of graphene in G/SBSMA was determined to be 2%, the optimal dosage of crumb rubber in CR/SBSMA was determined to be 15%, and the optimal dosage of C9 petroleum resin in PR/SBSMA was determined to be 5%.

(3) Based on the theory of asphalt surface free energy, after aging, asphalt generated more large molecular substances internally, which reduced the number of freely movable molecules and enhanced the nonpolar component of the asphalt. As a result, the surface energy tended to increase. The intercalated structure of graphene, the flocculent structure of rubber powder, and the aromatic substances of resin can all improve the adhesive properties of SBS modified asphalt before and after aging.

(4) As aging progressed, the cohesive energy and exfoliated work of asphalt continuously increased, while the adhesive energy decreased. Simultaneously, the decrease in the energy ratio parameter indicated the increasing activity of the asphalt-aggregate biphasic system, thereby revealing the mechanism of strength attenuation at the asphalt–aggregate interface when exposed to water and aging.

(5) Considering the influences of material composition, aging gradient, and environmental factors, resin-based anti-aging agents exhibited the most noticeable improvement in asphalt adhesion performance. However, graphene significantly enhanced the water stability performance of aged asphalt during the aging stage, resulting in a more significant improvement in adhesive state stability.

## Figures and Tables

**Figure 1 materials-16-04881-f001:**
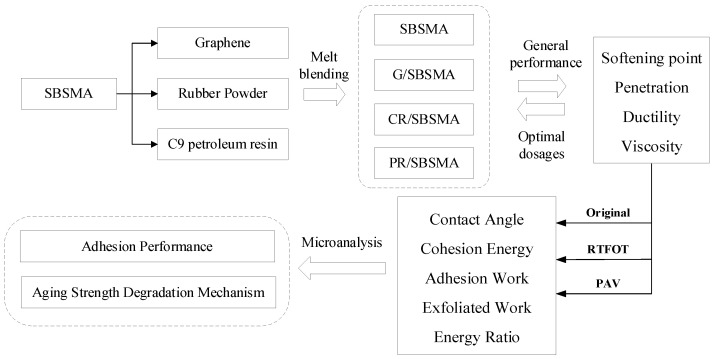
Technology methodology.

**Figure 2 materials-16-04881-f002:**
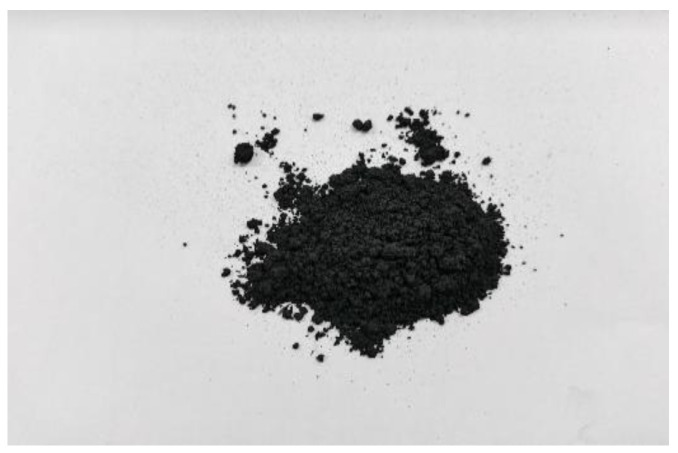
Graphene powder.

**Figure 3 materials-16-04881-f003:**
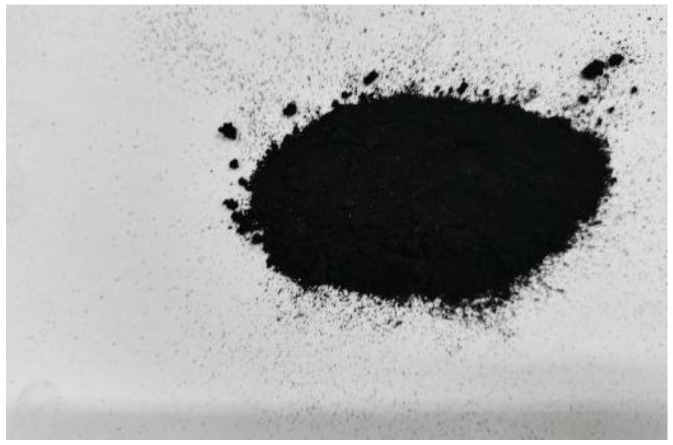
Rubber powder.

**Figure 4 materials-16-04881-f004:**
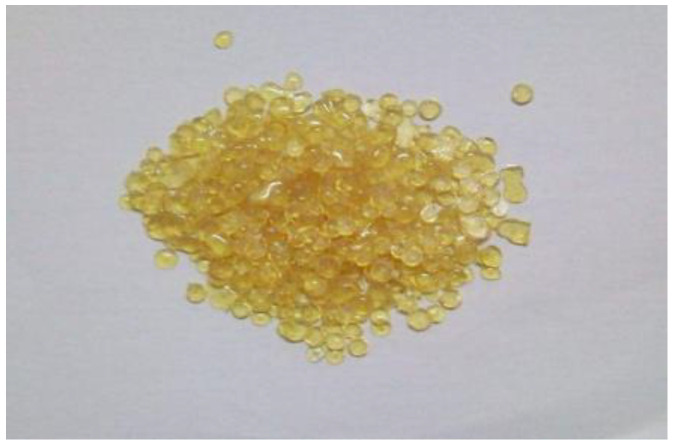
C9 petroleum resin.

**Figure 5 materials-16-04881-f005:**
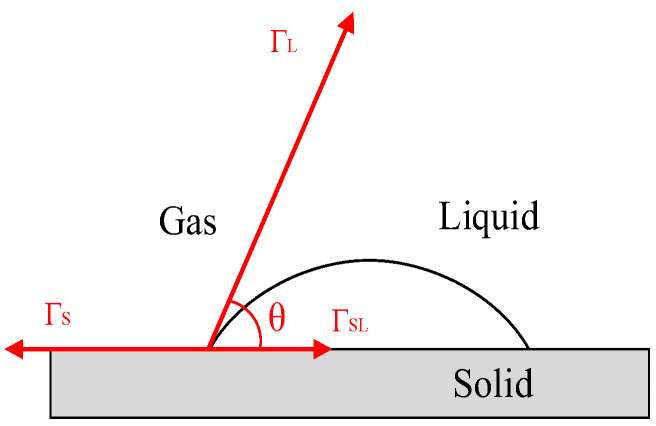
Schematic diagram of contact angle.

**Figure 6 materials-16-04881-f006:**
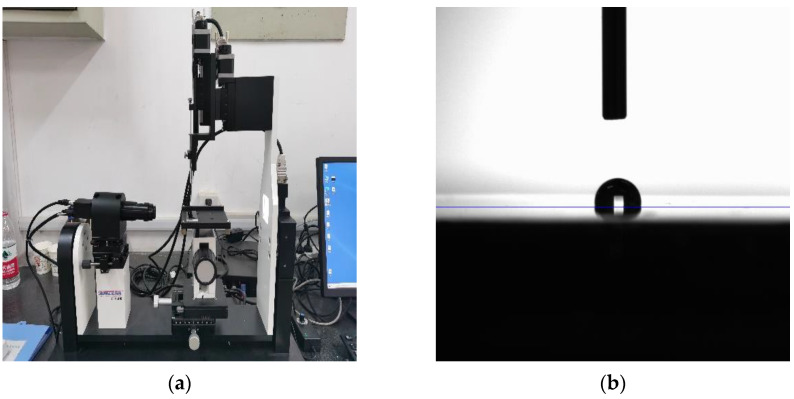
Contact angle test: (**a**) contact angle instrument; (**b**) baseline setting.

**Figure 7 materials-16-04881-f007:**
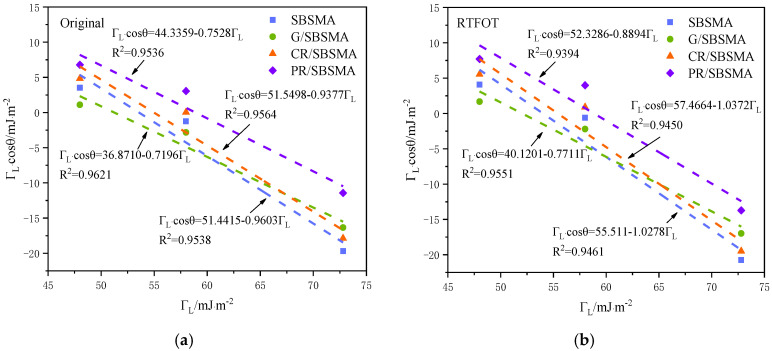
Fitting results of asphalt contact angle: (**a**) original condition; (**b**) short-term aging condition; (**c**) long-term aging condition.

**Figure 8 materials-16-04881-f008:**
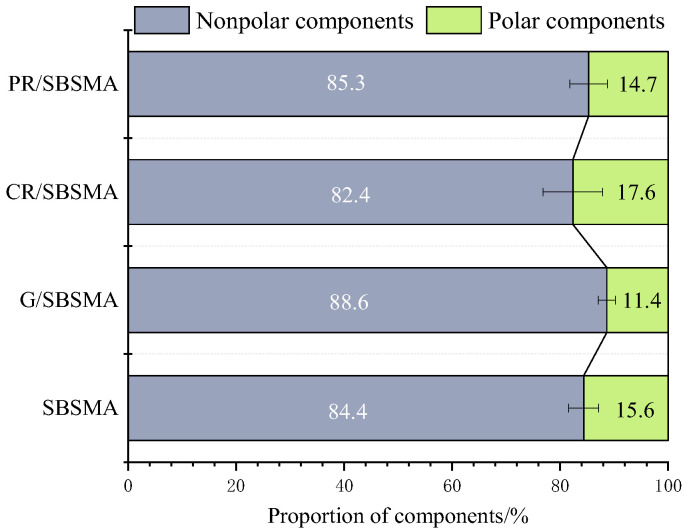
Proportion of asphalt components.

**Figure 9 materials-16-04881-f009:**
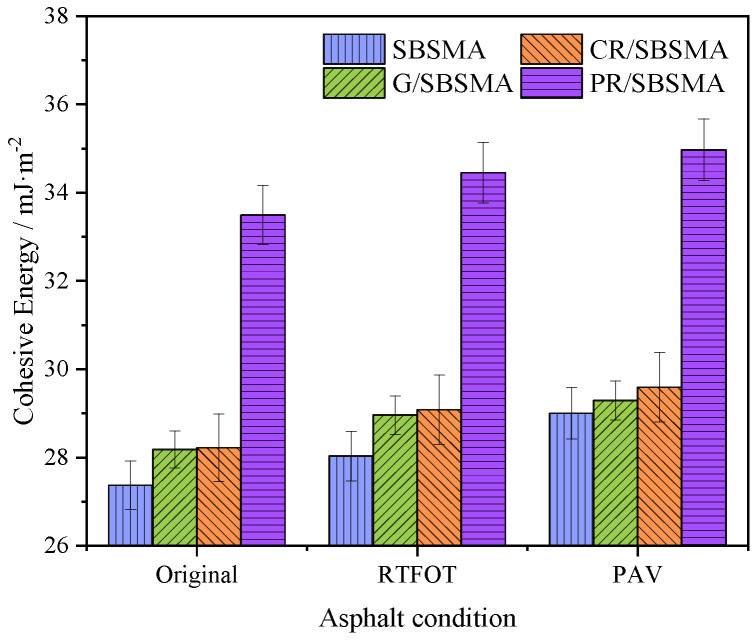
Cohesive energy of asphalt.

**Figure 10 materials-16-04881-f010:**
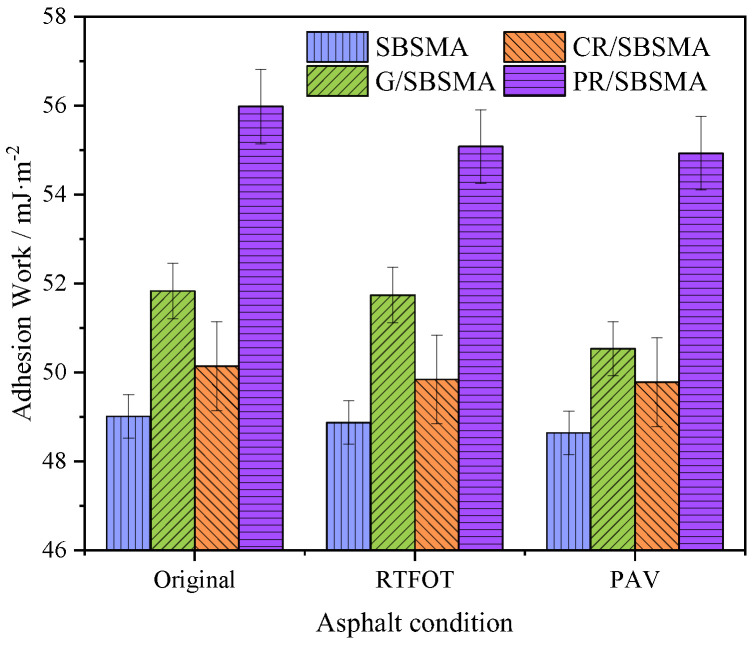
Adhesive work of asphalt.

**Figure 11 materials-16-04881-f011:**
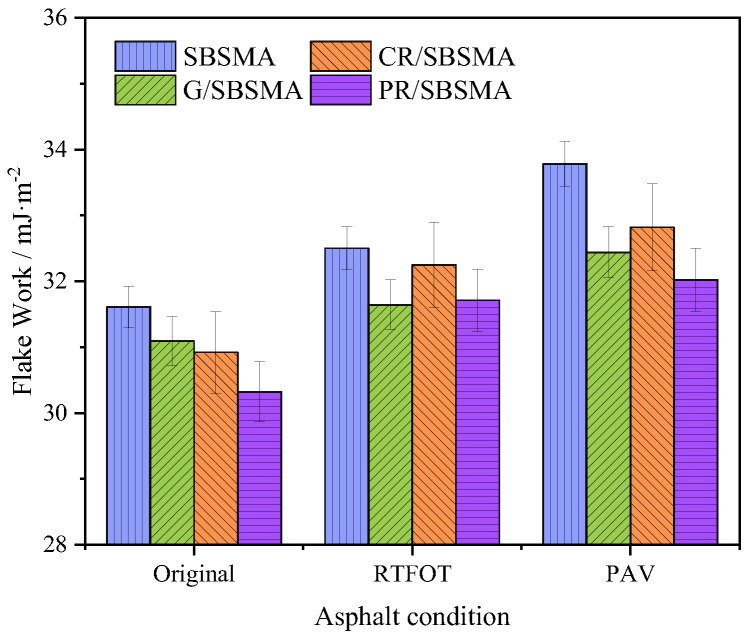
Exfoliated work of asphalt.

**Figure 12 materials-16-04881-f012:**
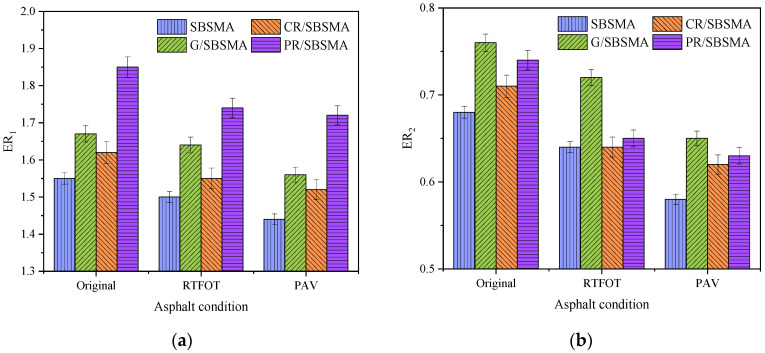
Energy ratios: (**a**) ER_1_; (**b**) ER_2_.

**Table 1 materials-16-04881-t001:** Technical performance index of SBSMA.

Index	Detection Results	Technical Requirement	Test Method
25 °C Penetration/0.1 mm	52.5	40–60	T0604
Penetration index PI	0.15	≮0	T0604
5 °C Ductility/cm	31.6	≮20	T0606
Softening point/°C	72.8	≮60	T0620
135 °C Viscosity/(Pa·s)	1.53	≯3	T0605
25 °C Elastic recovery/%	83	≮75	T0611
15 °C Density/(g·cm^−3^)	1.038	Actual observation record	T0603

**Table 2 materials-16-04881-t002:** Main parameters of graphene.

Appearance	Number of Layers /Layers	Thickness /nm	Specific Surface Area/(m^2^·g^−1^)	Lamellar Diameter/μm	Oxygen Content/%	Sulphur Content/%
Black powder	6–10	3.4–8	100–300	5~50	0.5	0.5

**Table 3 materials-16-04881-t003:** Main parameters of rubber powder.

Appearance	Relative Density	Moisture Content/%	Rubber Hydrocarbon Content/%	Acetone Extract/%	Carbon Black Mass Fraction/%
Black powder	1.16	0.55	58.2	7.9	32.8

**Table 4 materials-16-04881-t004:** Main parameters of C9 petroleum resin.

Appearance	Relative Density	Refractive Index	Acid Value	Softening Point/°C	Wax Fog Point/°C
Yellow granules	0.98	1.512	≤0.5	110	78

**Table 5 materials-16-04881-t005:** Test liquid surface energy parameters/(mJ·m^−2^).

Type of Test Solution	Γ*_L_*	$\Gamma_{L}^{LW}$	$\Gamma_{L}^{AB}$	$\Gamma_{L}^{+}$	$\Gamma_{L}^{-}$
Formamide	58.0	39.0	19.0	2.28	39.6
Glycol	48.0	29.0	19.0	1.92	47.0
Distilled water	72.8	21.8	51.0	25.5	25.5

**Table 6 materials-16-04881-t006:** Surface energy parameters of aggregates/(mJ·m^−2^).

Type of Aggregate	Γ*_S_*	$\Gamma_{S}^{LW}$	$\Gamma_{S}^{AB}$	$\Gamma_{S}^{+}$	$\Gamma_{S}^{-}$
Limestone	46.24	28.67	17.57	9.23	8.36

**Table 7 materials-16-04881-t007:** G/SBSMA test data.

Index	Graphene Content						Technical Requirement
	0%	0.5%	1%	1.5%	2%	2.5%	
Softening point/°C	72.8	74.1	75.3	76.3	77.2	77.9	≮60
25 °C Penetration/0.1 mm	52.5	51.2	50	48.8	48.2	46.8	40~60
5 °C Ductility/cm	31.6	32.1	32.4	32.7	32.8	31.5	≮20
135 °C Viscosity/(Pa·s)	1.53	1.68	1.71	1.82	1.92	1.99	≯3

**Table 8 materials-16-04881-t008:** CR/SBSMA test data.

Index	Rubber Powder Content						Technical Requirement
	0%	10%	12.5%	15%	17.5%	20%	
Softening point/°C	72.8	70.2	71.8	72.8	73.2	73.5	≮60
25 °C Penetration/0.1 mm	52.5	55.2	54.5	53.5	50.6	45.3	40~60
5 °C Ductility/cm	31.6	33.1	33.7	34.2	34.5	33	≮20
135 °C Viscosity/(Pa·s)	1.53	1.91	2.06	2.29	2.38	2.46	≯3

**Table 9 materials-16-04881-t009:** PR/SBSMA test results.

Index	Petroleum Resin Content						Technical Requirement
	0%	3%	4%	5%	6%	7%	
Softening point/°C	72.8	75.3	78.2	80	81.5	82.2	≮60
25 °C Penetration/0.1 mm	52.5	49.9	49.2	48.7	48.3	48.1	40~60
5 °C Ductility/cm	31.6	34.6	35.9	37.5	36.8	35.5	≮20
135 °C Viscosity/(Pa·s)	1.53	2.05	2.27	2.53	2.71	2.85	≯3

**Table 10 materials-16-04881-t010:** Asphalt contact angle data.

Asphalt Type	Aging Condition	Formamide /°	Coefficient of Variation/%	Glycol/°	Coefficient of Variation/%	Distilled Water/°	Coefficient of Variation/%
SBSMA	Original	91.22	1.88	85.77	1.55	105.69	1.79
	RTFOT	90.58	1.52	85.12	2.41	106.54	1.46
	PAV	90.15	2.23	84.86	1.95	107.78	1.55
G/SBSMA	Original	92.79	1.44	88.68	2.63	102.97	0.88
	RTFOT	92.18	2.51	87.99	3.62	103.49	1.64
	PAV	91.25	1.38	86.57	2.44	105.13	1.94
CR/SBSMA	Original	89.97	3.76	84.22	2.85	104.21	3.23
	RTFOT	89.12	2.92	83.37	3.65	105.53	2.08
	PAV	87.65	3.55	81.51	3.44	106.05	3.11
PR/SBSMA	Original	86.99	3.05	81.88	2.56	99.03	2.15
	RTFOT	86.07	2.87	80.75	2.08	100.86	1.88
	PAV	85.36	2.55	79.82	1.92	101.23	2.07

## Data Availability

Not applicable.

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
