# Peer review of "Study on Adhesion Performance and Aging Strength Degradation Mechanism of SBS Modified Asphalt with Different Anti-Aging Additive"

_materials, 2023, doi:10.3390/ma16134881_

Round 1
Reviewer 1 Report
This study is aimed to assess the efficiency of three anti-aging additive on aging performance of asphalt binder. The following questions should be clarified and some points should be taken into account.
-Some research questions have been answered in the provided introduction. Therefore, I think authors must be able to justify the novelty of their research in a separate section like 'goals and objectives'. There you should also describe the research limitations and contribution to the body of the literature.
- The conventional tests (Penetration, softening and etc.) were used to determine the optimum content of the agents. These tests are considered as empirical tests used only for assessing unmodified bitumen. Authors are advised to justify why these tests were used to determine the best contents. Also, since this study is focused on evaluating the aging performance of bitumen, then why the addition of these agents was done on unaged binder
- This study found that The optimal dosages of graphene, crumb rubber, and C9 petroleum resin were determined to be 2%, 16 15%, and 5% respectively. Authors should explain how these values were obtained as it is not clear in section "2.2.3 Conventional performance evaluation line 204" how these values were obtained. Usually the conventional tests used to find the optimum value of additive that recover the original properties of aged binder. For example, in the study the penetration value for unaged SBS binder is 52.5, however after the addition of agents (Graphene and resin), the penetration of modified asphalts (MA) were decreased considerably (MA become very hard) and increased for rubber (MA become very soft). Please look at this paper (https://doi.org/10.1016/j.conbuildmat.2022.127026) read section "6.1. Conventional tests" also used as references (This study proposed that the percentage of anti-agents that restore the penetration of aged binder can correspondingly fulfill the Superpave criterion for anti-agent content)
- The Introduction is very long. It must be shorter and more straightforward. In addition, knowledge gap should be clarified clearer. The rest of the introduction can be moved to a separate section as literature review.
- There are obvious grammatical errors. For this reason, the integrity of the piece is shaken (Passive use). Pls avoid using "he or she" as shown in line 50"
- Abstract is quite long, Pls shorten by including the main test and results, also show the problem statement more clearly.
- Pls add SBS binder in fig1 since it was tested.
- For section "2.1.1. Asphalt 141" Pls use more straightforward to describe the binder used in this study such as source and whether this type of binder is widely used in your country.
- Delete fig 2 as these tests are very basic.
- Pls use more quality pics for Figs 3 to 5.
- Lines 177-178, pls use rpm instead of kr
- pls move the section "2.2.3 Conventional performance evaluation 204" to section of results and in the method section, authors can state these test and their standards.
- In section of 2.2.3" Authors are discussed the results by stating some fact, Authors should add references for these facts such as in line 246 to 249
- Lines 361 to 363, add the figure that used to describe these findings
- Figure 7 was not cited in the text.
- Line 380 to382, Authors are suggested to add references for this explanation (Authors could use " https://doi.org/10.3390/su13126523
" as it stated here in section "4.2.2. SARA fractionation" that "As age progresses, aromatics are thought to be transformed first into resins, then into asphaltenes"
- Line 450, pls delete "The experimental results were as follows."
- Pls add more updated and related references.
- There are several sentences are very long and hard to understand.
- Pls check the tenses as in the results section, the future tense was used. Authors should either use present tense for facts and past tense for what it was obtained.
- The whole text needs to have a proofreading by a native-speaker expert in field
Author Response
Response Letter
Manuscript ID: materials-2468955
Title: Study on Adhesion Performance and Aging Strength Degradation Mechanism of SBS Modified Asphalt with Different Anti-aging Additive
Dear Reviewer,
Firstly, thank you for your kind patience. My team really appreciate your careful review and constructive comments concerning this paper. These comments are all valuable and helpful for improving this paper. All the authors have carefully discussed about all these comments.
we have accepted and revised as recommended in this revised manuscript. All modifications have been marked in red font. The point-by-point responses are provided below. We hope that the revision is acceptable, and your favorable consideration of our manuscript is greatly appreciated.
If you have any questions regarding this version of manuscript, please feel free to contact me. We are looking forward to hearing from you soon.
Best regards,
Chuanyi Zhuang and Yan Hao
Shandong Jiaotong University, Jinan, Shandong, China
Email: 204068@sdjtu.edu.cn and sdjthaoyan@163.com
Response to the reviewers’ comments:
This study is aimed to assess the efficiency of three anti-aging additive on aging performance of asphalt binder. The following questions should be clarified and some points should be taken into account.
Comments 1:
Some research questions have been answered in the provided introduction. Therefore, I think authors must be able to justify the novelty of their research in a separate section like 'goals and objectives'. There you should also describe the research limitations and contribution to the body of the literature.
Response 1:
Thank you for your comments on the suggestions. I have restructured the introduction and highlighted the limitations of the current study as well as the novelty of the research in this paper. I hope that it meets your requirements. The supplements are located in lines 95-110.
Lines 207-209:
Currently, research on anti-aging of asphalt mainly focuses on macroscopic properties such as high temperature, low temperature, water stability, and fatigue, with less emphasis on the study of microcosmic adhesion strength degradation mechanisms before and after aging. Additionally, due to the differences in material characteristics, different anti-aging agents exhibit varying performance improvements when compounded with SBS, and the degradation patterns of adhesion performance are inconsistent. Therefore, this paper selects three popular anti-aging modifiers, namely, nano graphene, rubber powder and C9 petroleum resin, and uses the melt blending process to mix them with SBS modified asphalt to prepare composite modified asphalt. The adhesion index of the asphalt and aggregate interface is assessed using the Lying Drop Method based on the surface free energy theory. Quantitative analysis is performed on the variations in adhesion performance between asphalt and aggregate before and after aging , revealing the mechanism of asphalt-aggregate adhesion degradation and strength failure after aging. By studying the effect of different anti-aging agents on the adhesion performance of asphalt, it provides a basis for the comparison and selection of asphalt binders such as asphalt pavement, bridge deck pavement and ultra-thin overlay.
Comments 2:
The conventional tests (Penetration, softening and etc.) were used to determine the optimum content of the agents. These tests are considered as empirical tests used only for assessing unmodified bitumen. Authors are advised to justify why these tests were used to determine the best contents. Also, since this study is focused on evaluating the aging performance of bitumen, then why the addition of these agents was done on unaged binder.
Response 2:
The reason for determining the optimum dosing of anti-ageing agents based on conventional tests is that the tests are more workable and reproducible. In China this evaluation method is often applied to modified asphalt as well. In addition, the material properties of the asphalt and the modifier are listed in the section 2.1 to allow the researcher to carry out repeatable tests.
Comments 3:
This study found that The optimal dosages of graphene, crumb rubber, and C9 petroleum resin were determined to be 2%, 16 15%, and 5% respectively. Authors should explain how these values were obtained as it is not clear in section "2.2.3 Conventional performance evaluation line 204" how these values were obtained. Usually the conventional tests used to find the optimum value of additive that recover the original properties of aged binder. For example, in the study the penetration value for unaged SBS binder is 52.5, however after the addition of agents (Graphene and resin), the penetration of modified asphalts (MA) were decreased considerably (MA become very hard) and increased for rubber (MA become very soft). Please look at this paper
(https://doi.org/10.1016/j.conbuildmat.2022.127026) read section "6.1. Conventional tests" also used as references (This study proposed that the percentage of anti-agents that restore the penetration of aged binder can correspondingly fulfill the Superpave criterion for anti-agent content)
Response 3:
Due to a previous oversight, the literature was not added when the different modifier dosing levels were initially determined. This has been added in the revised version and the literature you have listed has been read as suggested. The supplements are located in lines 143-164.
Lines 143-164:
With the purpose of giving full play to the modification effect of the three anti-aging agents, from the perspective of the shear process and economy, three kinds of composite-modified asphalt were prepared by the melt blending method.
(1) Graphene / SBS composite modified asphalt (G/SBSMA)
The G / SBSMA was prepared by adding 0.5%, 1.0%, 1.5%, 2.0% and 2.5% graphene additive [23]. The temperature was set at 170℃, and graphene was added to asphalt at a rate of 2kpm in a small number of multiple times within 30 min. The G/SBSMA composite modified asphalt was obtained by adjusting the speed to 3kpm and shearing for 30 min.
(2) Crumb rubber / SBS composite modified asphalt (CR/SBSMA)
The blending mass ratios were set to 10%, 12.5%, 15%, 17.5% and 20%, respectively, and the powder was dried in advance [24]. The rubber powder was added to the asphalt in a small amount and many times at 180℃ shear temperature and 1.5kpm shear rate for 30 min. The shear temperature was kept constant, and the shear rate was adjusted to 4kpm for 60 min. After the shear was completed, the asphalt was placed in an oven at 170℃ for 30 min to swell and develop, and finally CR/SBSMA composite modified asphalt was prepared.
(3) Petroleum resin / SBS composite modified asphalt (PR/SBSMA)
The mixing amount is 3%, 4%, 5%, 6% and 7% [22]. At the shear temperature of 170℃and the shear rate of 2kpm, C9 petroleum resin was added in a small amount and multiple times for 30 min. PR/SBSMA composite modified asphalt was prepared by increasing the shear rate to 4kpm and high-speed shearing for 30 min to fully combine the resin and asphalt.
Comments 4:
The Introduction is very long. It must be shorter and more straightforward. In addition, knowledge gap should be clarified clearer. The rest of the introduction can be moved to a separate section as literature review.
Response 4:
We were grateful for the suggestion and streamlined the introduction. The supplements are located in lines 30-113.
Lines 30-113:
Inadequate adhesive between asphalt and aggregate and poor aging resistance of asphalt can readily result in cracks, looseness, potholes, and other asphalt pavement diseases. Therefore, improving the adhesive between asphalt and aggregate and enhancing the long-term aging resistance of asphalt are the goals pursued by road workers. These goals are also core elements related to the long-term durability of road surfaces [1]. Aging modifiers are frequently utilized in asphalt and its mixtures to increase pavement longevity and maintain high service levels [2]. Among them, nano graphene, rubber powder, petroleum resin and SBS modified asphalt have good compatibility and excellent performance. They are widely used modified asphalt materials.
The unique layered structure of graphene endows it with extremely strong surface activity, which can improve the mechanical strength of asphalt mixtures when physically blended with asphalt. It can also hinder the diffusion and penetration of oxygen in asphalt, potentiate the thermal stability and barrier performance, and achieve the goal of delaying aging. Li [3] pointed out that a 1% content of graphene can reduce the temperature receptivity of the base asphalt and improve the rheological properties of the asphalt before and after aging. Liu [4] proposed that a 0.4% content of graphene can reduce the phase angle of SBS modified asphalt and increase its resistance to permanent deformation. Additionally, graphene can reduce the rate of change of the carbonyl index and sulfoxide index after PAV aging. Dong [5] added 2%, 3% and 4% graphene into the base asphalt. Using the linear amplitude scanning (LAS) test, and pointed out that the aging resistance of the modified asphalt can be improved when the content of graphene is not more than 3%. The aging resistance effect is more pronounced under the condition of large strain. Zeng [6] established the molecular model of the asphalt system with graphite oxide, and confirmed that graphite oxide can hinder the movement and volatilization of saturated hydrocarbons to a certain extent, thus enhancing the aging resistance of asphalt, based on the Density functional theory (DFT) theory. Wang [7] also pointed out that graphite oxide in favor of the crystallization of SBSMA, and changes in oxygen containing groups have a positive function on the aging resistance and fatigue endurance of asphalt. Liu [8] highlighted that graphite oxide can not only improve the water stability of the mixture after aging, but also slow down the damage to low-temperature performance.
Waste tire rubber powder, abbreviated as rubber powder, has been widely used in asphalt mixtures to improve the durability of asphalt pavement [9]. However, the modification of rubber powder with asphalt needs to be carried out at higher temperatures and the environmental impact of the fumes generated during the process cannot be ignored [10,11]. Suo [12] mixed 40-mesh rubber powder with 5%, 10%, 15% and 20% of SBS modifier, and found that the anti-aging effect of composite modified asphalt was better than that of SBS modified asphalt. Jamal [13] modified the base asphalt with 30-mesh, 22.5% high content rubber powder. After laboratory simulated long-term aging and UV aging tests, it was found that the aging index did not change significantly. Using PG grading indicators, Li [14] confirmed that an appropriate amount of rubber powder can improve the low-temperature performance and fatigue performance of asphalt after aging. Xiang [15] studied the micro composition and structural changes of 40 to 60 mesh rubber powder and SBS before and after thermal oxygen aging, and pointed out that the degradation products of the modifier after aging will react with the secondary components of asphalt, which will lead to a decrease in aromatic and resin content and an increase in asphaltene content. The increase in the number of macromolecules, it will contribute to improve micro and macro mechanical properties. Ren [16] established a molecular model of rubber powder modified asphalt using molecular dynamics. Rubber powder can improve the flowability of asphalt molecules, thereby reducing the influence of aging on the low-temperature performance of asphalt.
Modified asphalt with high viscosity has excellent adhesion and elastic recovery ability. The modifiers used to prepare high viscosity asphalt are not unique. Currently, they are mainly divided into rubber, resin, and thermoplastic elastomer types, all of which can be added separately or in a composite form to increase the viscosity of asphalt, in order to achieve the standard of high viscosity asphalt with a dynamic viscosity of no less than 20000 Pa·s at 60℃ [17,18]. Sun [19] found that the phase distribution of asphalt with high viscosity is more uniform and is little affected by the combined effects of oxidation induced hardening and degradation induced softening, demonstrating better aging resistance. Qiu [20] proposed that rubber based high viscosity agents have the best storage stability and temperature sensitivity through the microscopic and macroscopic indicators of asphalt under different aging states. Shi [21] used C9 petroleum resin and SBS to prepare high viscosity and high elasticity asphalt modifier SPR, and found that the economic benefit and anti-aging effect were better. Yang [22] pointed out that aging reduces the strength of functional groups of high viscosity asphalt, but the aging resistance is significantly better than that of base asphalt and SBS modified asphalt.
Currently, research on anti-aging of asphalt mainly focuses on macroscopic properties such as high temperature, low temperature, water stability, and fatigue, with less emphasis on the study of microcosmic adhesion strength degradation mechanisms before and after aging. Additionally, due to the differences in material characteristics, different anti-aging agents exhibit varying performance improvements when compounded with SBS, and the degradation patterns of adhesion performance are inconsistent. Therefore, this paper selects three popular anti-aging modifiers, namely, nano graphene, rubber powder and C9 petroleum resin, and uses the melt blending process to mix them with SBS modified asphalt to prepare composite modified asphalt. The adhesion index of the asphalt and aggregate interface is assessed using the Lying Drop Method based on the surface free energy theory. Quantitative analysis is performed on the variations in adhesion performance between asphalt and aggregate before and after aging, revealing the mechanism of asphalt-aggregate adhesion degradation and strength failure after aging. By studying the effect of different anti-aging agents on the adhesion performance of asphalt, it provides a basis for the comparison and selection of asphalt binders such as asphalt pavement, bridge deck pavement and ultra-thin overlay. The research methodology is shown in Figure 1.
Figure 1. Technology methodology.
Comments 5:
There are obvious grammatical errors. For this reason, the integrity of the piece is shaken (Passive use). Pls avoid using "he or she" as shown in line 50"
Response 5:
We have fixed the problem here. See Response 4 for the revised introductory section.
Comments 6:
Abstract is quite long, Pls shorten by including the main test and results, also show the problem statement more clearly.
Response 6:
We have made a brief deletion to the summary section, which is located on lines 10-25 after the change.
Line 10-25:
Abstract: After aging, the adhesiveness of asphalt deteriorates, leading to a reduction in the durability of asphalt mixtures and affecting the service life of asphalt pavements. To enhance the anti-aging performance of asphalt, this study employed the method of melt blending to prepare three types of modified asphalt: graphene/SBS-modified asphalt (G/SBSMA), crumb rubber/SBS-modified asphalt (CR/SBSMA), and petroleum resin/SBS-modified asphalt (PR/SBSMA). Different dosages of the three types of modified asphalt were tested for changes in conventional performance indicators. The optimal dosages of graphene, crumb rubber, and C9 petroleum resin were determined to be 2%, 15%, and 5% respectively. Based on the theory of surface free energy the effects of anti-aging agents on the microscopic properties of SBS-modified asphalt before and after aging were analyzed using the three-liquid method. The mechanisms of strength attenuation at the asphalt-aggregate interface under water exposure and aging were revealed. The results showed that with the increase of aging gradient, the asphalt-aggregate biphasic system became more active. The cohesive energy and peel energy of SBS-modified asphalt increased continuously, while the adhesive energy decreased continuously, leading to a decrease in the energy ratio parameter. Resin-based anti-aging agents exhibited the most significant improvement in asphalt adhesion performance, while graphene demonstrated a more stable enhancement in asphalt's water stability during the aging stage.
Comments 7:
Pls add SBS binder in fig1 since it was tested.
Response 7:
SBSMA has been added to the Figure 1 technology methodology.
Comments 8:
For section "2.1.1. Asphalt 141" Pls use more straightforward to describe the binder used in this study such as source and whether this type of binder is widely used in your country.
Response 8:
Rubber powder is already widely used in road construction in China, while research on petroleum resins and graphene is just in its infancy.
Comments 9:
Delete fig 2 as these tests are very basic
Response 9:
Figure 2 has been removed.
Comments 10:
Pls use more quality pics for Figs 3 to 5.
Response 10:
Both the graphene and the rubber powder have a black appearance, but due to personal travel, a higher quality image is not available at this time and at short notice.
Comments 11:
Lines 177-178, pls use rpm instead of kr
Response 11:
For the use of rpm units I have made changes as you suggested.
Comments 12:
Pls move the section "2.2.3 Conventional performance evaluation 204" to section of results and in the method section, authors can state these test and their standards.
Response 12:
I have moved that section to section 3.
The test method was carried out in accordance with the 'Standard Test Methods of Bitumen and Bituminous Mixtures for Highway Engineering' (JTG E20-2011).
Comments 13:
In section of 2.2.3" Authors are discussed the results by stating some fact, Authors should add references for these facts such as in line 246 to 249
Response 13:
For the content here we have added literature citations. For example, document 29 is the actual situation described.
Comments 14:
Lines 361 to 363, add the figure that used to describe these findings
Response 14:
For the lack of data to support the description here, the data has been listed in the revised version in the form of Table 10. The supplements are located in line 351.
Table 10. Asphalt contact Angle data
Asphalt type |
Aging condition |
Formamide / ° |
Coefficient of variation / % |
Glycol / ° |
Coefficient of variation / % |
Distilled water / ° |
Coefficient of variation / % |
SBSMA |
Original |
91.22 |
1.88 |
85.77 |
1.55 |
105.69 |
1.79 |
RTFOT |
90.58 |
1.52 |
85.12 |
2.41 |
106.54 |
1.46 |
|
PAV |
90.15 |
2.23 |
84.86 |
1.95 |
107.78 |
1.55 |
|
G/SBSMA |
Original |
92.79 |
1.44 |
88.68 |
2.63 |
102.97 |
0.88 |
RTFOT |
92.18 |
2.51 |
87.99 |
3.62 |
103.49 |
1.64 |
|
PAV |
91.25 |
1.38 |
86.57 |
2.44 |
105.13 |
1.94 |
|
CR/SBSMA |
Original |
89.97 |
3.76 |
84.22 |
2.85 |
104.21 |
3.23 |
RTFOT |
89.12 |
2.92 |
83.37 |
3.65 |
105.53 |
2.08 |
|
PAV |
87.65 |
3.55 |
81.51 |
3.44 |
106.05 |
3.11 |
|
PR/SBSMA |
Original |
86.99 |
3.05 |
81.88 |
2.56 |
99.03 |
2.15 |
RTFOT |
86.07 |
2.87 |
80.75 |
2.08 |
100.86 |
1.88 |
|
PAV |
85.36 |
2.55 |
79.82 |
1.92 |
101.23 |
2.07 |
Comments 15:
Figure 7 was not cited in the text.
Response 15:
We have cited a reference to Figure 7 in the text. As a result of the deletion of Figure 2, the previous Figure 7 was updated to Figure 6, with the citation located in lines 349-350.
Comments 16:
Line 380 to382, Authors are suggested to add references for this explanation
(Authors could use " https://doi.org/10.3390/su13126523
" as it stated here in section "4.2.2. SARA fractionation" that "As age progresses, aromatics are thought to be transformed first into resins, then into asphaltenes"
Response 16:
Documentation has been added here, see the introduction to document 30 in line 366.
Comments 17:
Line 450, pls delete "The experimental results were as follows."
Response 17:
This has been removed.
Comments 18:
Pls add more updated and related references.
Response 18:
The References section has been removed and updated. The updated references are shown below.
- Peng, Q.; Hao, Z.H.; Tan, Y.Q. Influence analysis of UV-aging on the properties of high elastic modified asphalt and mixture. Highw. Trans. 2020, 36, 32-37. DOI:10.13607/j.cnki.gljt.2020.05.006.
- Arsalan, R.; Imran, K.; Faisal, T.R. Evaluation of Moisture Damage Potential in Hot Mix Asphalt Using Polymeric Aggregate Treatment. Materials. 2022, 15, 5437.
- Li, X.; Wang, Y.M.; Wu, Y.L. Properties and modification mechanism of asphalt with graphene as modifier. Build. Mater. 2021, 272, 121919.
- Liu, Z.H.; Jiang, W.; Wang, L. Effect of graphene on rheological properties of SBS modified bitumen. Highway. 2022, 67, 76-82.
- Dong, X.Y. Research on performance of nano-graphene modified asphalt. Master’s Thesis, Southeast University, Nanjing, China, 2021.
- Zeng, Q.; Liu, Y.R.; Liu, Q.C. Preparation and modification mechanism analysis of graphene oxide modified asphalts. Build. Mater. 2020, 238, 117706.
- Wang, R.R.; Yue, M.J.; Xiong, Y.C. Experimental study on mechanism, aging, rheology and fatigue performance of carbon nanomaterial/SBS-modified asphalt binders. Build. Mater. 2021, 268, 121189.
- Liu, K.F.; Zhu, J.C.; Wu, C.F. Experimental study on aging resistance of graphene oxide modified asphalt and its mixture. Highway. 2020, 65, 225-230.
- Bolakar, T.H. Production and characterisation of waste tire pyrolytic oil - Investigating physical and rheological behaviour of pyrolytic oil modified asphalt binder. Heliyon. 2023, 9, e12851.
- Pouranian, M.R.; Notani, M.A.; Tabesh, M.T. Rheological and environmental characteristics of crumb rubber asphalt binders containing non-foaming warm mix asphalt additives. Build. Mater. 2020, 238, 117707.
- Fernández, I.R.; Cavalli, M.C.; Poulikakos, L. Recyclability of Asphalt Mixtures with Crumb Rubber Incorporated by Dry Process: A Laboratory Investigation. Materials. 2020, 13, 2870.
- Suo, L.J. Preparation and properties of rubber powder/SBS composite modified asphalt binder. Mater. 2022, 53, 6224-6229.
- Jamal, M.; Giustozzi, F. Enhancing the asphalt binder's performance against oxidative ageing and solar radiations by incorporating rubber from waste tyres. Build. Mater. 2022, 350, 128803.
- Li, S.P.; Li, Y.F.; Chen, J. Study on the rheological properties and performance of rubber and RET modified SBS asphalt before and after aging. Eng. 2017, 42, 112-120.
- Xiang, L.; Cheng, J.; Kang, S.J. Thermal oxidative aging mechanism of crumb rubber/SBS composite modified asphalt. Build. Mater. 2015, 75, 169-175.
- Ren, M.K.; Li, Y.M.; Cheng, P.F. Effect of modifier on low-temperature reversible aging behavior of asphalt binder and its morphology analysis. Build. Mater. 2022, 351, 128943.
- Akihiro, M.; Toshiro, J.; Takaaki, N. Construction and pavement properties after seven years in porous asphalt with long life. Build. Mater. 2014, 50, 401-413.
- Cao, D.W.; Lu, J.; Zhang, H.Y. Contrastive on performance of fully permeable asphalt pavement dedicated high-viscosity modified asphalt. Chang'an Univ. 2019,39, 21-28.
- Sun, G.Q.; Zhu, X.B.; Zhang, Q.Y. Oxidation and polymer degradation characteristics of high viscosity modified asphalts under various aging environments. Tot. Env. 2022, 813, 152601.
- Qiu, X.; He, L.Y.; Chen, J. Diagnostic of aging resistance and storage stability of high-performance modified asphalts under multi-scale conditions. Zhejiang Norm. Univ. 2023, 46, 169-178. DOI:10.16218/j.issn.1001-5051.2023.006.
- Shi, J.T.; Zhao, P.H.; Fan, W.Y. Facile preparation and application performance evaluation of SBS/C 9 petroleum resin blends as modifier for high viscosity asphalt. Build. Mater. 2020, 262, 120073.
- Yang, J.H.; Zhang, Z.Q.; Shi, J.R. Comparative analysis of thermal aging behavior and comprehensive performance of high viscosity asphalt (HVA) from cohesion, adhesion and rheology perspectives. Build. Mater. 2022, 317, 125982.
- Yerik A.; Aidos Y.; Assel, N. Characterization of asphalt bitumens and asphalt concretes modified with carbon powder. Case Stud Constr Mat. 2022, 17, e01554.
- Jihyeon, Y.; Mithil M.; IlHo, N. Evaluation of Effect of Thermoplastic Polyurethane (TPU) on Crumb Rubber Modified (CRM) Asphalt Binder. Materials. 2022, 15, 3824.
- Student, M.M.; Pokhmurs’ka, H.V.; Zadorozhna, K.R. Corrosion Resistance of VC–FeCr and VC–FeCrСо Coatings Obtained by Supersonic Gas-Flame Spraying. Sci. 2019, 54, 535-541.
- Reza, B.; Dariush, S.; Mahmoud, A. Correlation between bond strength and surface free energy parameters of asphalt binder-aggregate system. Build. Mater. 2021, 303, 124487.
- Naseri, Y.M.; Sarkar, A.; Hamedi, G.H. Application of the surface free energy method on the mechanism of low-temperature cracking of asphalt mixtures. Build. Mater. 2020, 268, 121194.
- Shu S.Q. Study on the improvement of adhesion between granite and asphalt and the road performance of mixture. Master’s Thesis, Shandong Jianzhu University, Jinan, China ,2022.
- Zhang, W.G.; Qiu, L.; Liu, J.P. Modification mechanism of C9 petroleum resin and its influence on SBS modified asphalt. Build. Mater. 2021, 306, 124740.
- Emiliano, P.; Edoardo, B. A Review on Bitumen Aging and Rejuvenation Chemistry: Processes, Materials and Analyses. Sustainability. 2021, 13, 6523.
- Ilyin, S.O.; Kostyuk, A.V.; Ignatenko, V.Y. The Effect of Tackifier on the Properties of Pressure-Sensitive Adhesives Based on Styrene–Butadiene–Styrene Rubber. J. Appl. Chem. 2018, 91, 1945-1956.
Comments 19:
Comments on the Quality of English Language:
- There are several sentences are very long and hard to understand.
- Pls check the tenses as in the results section, the future tense was used. Authors should either use present tense for facts and past tense for what it was obtained.
- The whole text needs to have a proofreading by a native-speaker expert in field
Response 19:
Formatting and grammatical errors have been corrected and we hope that the revised version will meet your requirements.

Reviewer 2 Report
The article "Study on Adhesion Performance and Aging Strength Degradation Mechanism of SBS Modified Asphalt with Different Anti-aging Additive" investigated the enhancement of the anti-aging performance of asphalt. The topic is relevant because of the need to extend the life of the asphalt concrete pavement in the conditions of a constant increase in traffic loads, traffic intensity and climatic factors, as well as the need to expand the functional properties of the pavement. The abstract is well written, complete and concise in various aspects. The keywords are complete and appropriate. The introduction is well written and finished. Materials and methods are clear and well explained. The results are easy to understand and comprehensive. All studied characteristics were presented in the figures. The conclusions are written clearly and understandably. The bibliography is formatted in accordance with the requirements of the journal and does not contain inappropriate references. But there are some comments on the article:
- At the end of the introduction, you need to indicate the purpose of the work.
- How much will adding such additives affect the price? Please indicate.
- There is not enough photomicrograph to see the distribution.
- Line 432: "G/SBSMA and PR/SBSMA have a higher proportion of nonpolar components, which enhances their adhesive properties with limestone due to the rich functional group structure." This statement requires confirmation by a reference. In general, C9 petroleum resin is a common tackifier widely used to increase the adhesive properties due to enhancing the rheological properties of an adhesive besides improving its wetting to glued surfaces (see 10.1134/S1070427218120054).
- Line 75: "Wang [12] tested the rheological properties of rubber powder SBS composite modified asphalt." The combined modification of road bitumen with rubber powder and SBS for improving rheological properties is also described in earlier work 10.1134/S1061933X1404005X.
Reviewer 3 Report
Dear Authors, Your manuscript deals with a very important topic - asphalt modification. In its current form it is not possible to accept it for publication yet. Please refer to my comments, suggestions, ambiguities and questions. 1. Please expand on the introduction: e.g. many researchers deal with the addition of rubber from tires to asphalt by changing their properties, e.g. Gawdzik (https://www.mdpi.com/1996-1944/13/21/4864; https://doi .org/10.1155/2018/8759549); Rodríguez-Fernández (https://www.mdpi.com/1996-1944/13/12/2870); Pouranian (https://doi.org/10.1016/j.conbuildmat.2019.117707) 2. Is the measurement of various parameters such as penetration or softening point made in accordance with any standard? 3. Please justify how the use of graphene in asphalt is economically compared to tire rubber or petrochemical resin. 4. Is there an economic sense and potential interest of the industry to implement any solution apart from the ecological management of tire waste? 5. It is difficult to compare the properties of bitumens with different additives when not using equal amounts. 6. The results for graphene are surprising: how is it possible that there is any amount of graphene in the sample compared to large amounts of bitumen. 7. Do I understand correctly that all modifications are based on physical modifications? 8. A chemical modification of rubber would give lower penetration (hardness, stiffness) and thus the material is soft - solutions known in the works of e.g. Gawdzik 9. Can the properties of tire rubber asphalt materials be affected by the soot present in them? 10. Is it possible that the (polar) resin is more easily degraded, e.g. by hydrolysis, than graphene or tire rubber? 11. Please extend the discussion of the results in relation to the solutions known in the literature/ 12. Please extend the conclusions. Best regards,
Round 2
Reviewer 1 Report
The authors have addressed some comments properly, however, some others were not responded
- for comments 2, the authors were asked to explain how the optimum values of agents were chosen and also why the conventional tests were done on unaged binder since this study is focussed on aged binder!!!!
- For comment 3, the authors were asked "This study found that The optimal dosages of graphene, crumb rubber, and C9 petroleum resin
were determined to be 2%, 16 15%, and 5% respectively. Authors should explain how these values
were obtained as it is not clear in section "2.2.3 Conventional performance evaluation line 204" how
these values were obtained. Usually the conventional tests used to find the optimum value of additive
that recover the original properties of aged binder. For example, in the study the penetration value
for unaged SBS binder is 52.5, however after the addition of agents (Graphene and resin), the
penetration of modified asphalts (MA) were decreased considerably (MA become very hard) and
increased for rubber (MA become very soft). Please look at this paper
(https://doi.org/10.1016/j.conbuildmat.2022.127026) read section "6.1. Conventional tests" also used
as references (This study proposed that the percentage of anti-agents that restore the penetration of
aged binder can correspondingly fulfill the Superpave criterion for anti-agent content)"
- pls respond to this more carefully as the authors have not revised to this comment properly.
- Pls change kpm to either "krpm" or add 000 to be rpm
Reviewer 3 Report
Dear Authors,
Thank you very much for sending detailed and exhaustive answers to my questions, suggestions.
Good luck - Accept
Author Response
We appreciate reviewer 3 for effort to review our manuscript, and positive feedback. The reviewer gives an accurate summary of our work and brings forward constructive questions.